# LOGITS REPLAY + MOCLIP: STABILIZED, LOW-COST POST-TRAINING WITH MINIMAL FORGETTING

## ABSTRACT

Large language models (LLMs) often face a trade-off in post-training: improvements on specialized domains frequently come at the expense of general capabilities. Existing solutions attempt to mitigate this tension via regularization, selective parameter updates, or data-centric replay, but each imposes significant costs in computation, data access, or adaptability. Recent work has shown that training signals can be compressed to subsets of logits without severe accuracy loss, suggesting a path toward efficient adaptation. However, naïve truncation destabilizzes optimization and exacerbates forgetting.

We introduce *Logits Replay + MoClip*, a two-stage framework that compresses supervision in the logit space and stabilizes optimization at the update level. In Stage 0, we record *dynamic Top-K* token subsets that cover a probability threshold, always including the gold label. In Stage 1, we replay these compact subsets to compute exact renormalized losses, avoiding full softmax computation and implicitly regularizing. To ensure stability, we design *MoClip*, an optimizer that caps gradient–momentum rotation and applies an $\arctan 2$-based rescaling of updates. Empirically, our method improves domain performance on Communication Technology (CT) and NL2SQL tasks while mitigating forgetting on general benchmarks (MMLU, BBH, GPQA, MATH), and reduces training cost by over 40%. Together, these contributions offer a scalable, architecture-agnostic path for domain adaptation of LLMs without sacrificing generalization.

## 1 INTRODUCTION

Fine-tuning large language models (LLMs) on domain-specific corpora often triggers notable degradation of general capabilities: gains in the new domain are offset by losses in general reasoning or knowledge(Lin et al., 2024; Kemker et al., 2018). This *see-saw effect* is well documented in continual learning and LLM post-training (Kirkpatrick et al., 2017; Aljundi et al., 2018; Zenke et al., 2017; Rebuffi et al., 2017; Javed & White, 2019; Mallya et al., 2018b; Ke et al., 2023; Hui et al., 2025). Existing remedies fall into three categories. *Regularization-based* approaches such as knowledge distillation (Hinton et al., 2015), Learning without Forgetting (Li & Hoiem, 2018; Mallya et al., 2018a; Aljundi et al., 2018; Yang et al., 2025), Classifier-Projection Regularization(Cha et al., 2021), and RecAdam (Chen et al., 2020) constrain updates toward the base model but reduce specialization. *Parameter-selective* methods, including MoFO (Chen et al., 2025), restrict updates to high-momentum weights to retain prior knowledge, yet sacrifice full plasticity. Similarly, parameter-efficient tuning methods like LoRA(Hu et al., 2021) have been shown to forget less than full fine-tuning but also underperform in-domain, acting as a form of implicit regularization(Biderman et al., 2024). *Data-centric* strategies, such as Baichuan4-Finance (Zhang et al., 2025) or SSR (Huang et al., 2024), preserve generality through replay or synthetic instance generation, but still require extra data or base model resources.

Recent work has explored optimizer-level stabilization and logit-level supervision. Torque-Aware Momentum (TAM) (Malviya et al., 2024) damps updates based on gradient–momentum angles, while AdaMuon (Si et al., 2025) adaptively rescales momentum. The Kimi K2 model (Team et al., 2025) introduced MuonClip and QK-Clip to prevent loss spikes in long-context training(Liu et al., 2025). These efforts highlight two promising directions: constraining optimization geometry and reusing model predictions. Yet, none unifies both perspectives in a lightweight, domain-agnostic framework.

In this work, we introduce *Logits Replay + MoClip*, a two-stage framework for efficient and stable LLM adaptation. **First**, Stage 0 records dynamic Top-$K$ logits per position, producing compact, entropy-adaptive candidate sets; Stage 1 replays these subsets to compute exact cross-entropy on restricted vocabularies, reducing cost and avoiding noisy gradients from low-probability tokens. The efficiency gain comes from computing Stage 1 loss only on the Top-$K$ vocabulary, eliminating nearly all softmax-related FLOPs. **Second**, to stabilize training under sparse supervision, we design *MoClip*, an optimizer that (i) caps gradient–momentum angles to enforce smooth update directions and (ii) applies $\texttt{arctan}\,2$-based rescaling to bound step sizes without relying on $\epsilon$. Compared to prior approaches, our method differs from MoFO by updating *all* parameters, from TAM by enforcing a hard geometry cap rather than soft damping, and from Baichuan4-Finance by avoiding external data replay. Extensive experiments show that Logits Replay + MoClip improves specialization on telecom QA and NL2SQL, preserves general reasoning performance, and reduces training cost by over 40%. **Overall**, this provides a balanced trade-off between plasticity and stability in post-training.

## 2 METHOD

Our training framework consists of two sequential stages: (1) Logits Replay Data Collection (Stage 0), and (2) Replay Fine-Tuning with MoClip (Stage 1). In Stage 0, we extract and save a compact, uncertainty-adaptive set of model predictions for each training example, which will serve as training targets in Stage 1. Stage 1 then fine-tunes the model on this reduced target space using our modified optimizer.

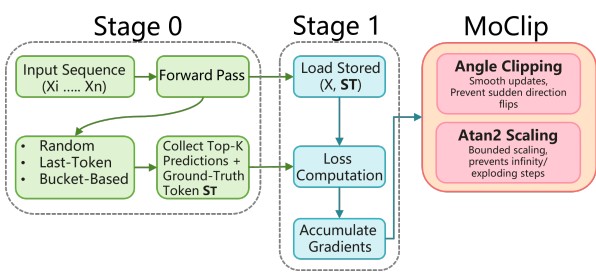

Figure 1: Overview of the Logits Replay + MoClip framework.

**Dynamic Top-$K$ selection (Stage 0).** Let $z_t \in \mathbb{R}^{|\mathcal{V}|}$ be the logits at position $t$ and $p_t = \mathrm{softmax}(z_t)$ the probabilities. Sort tokens by $p_t$ in descending order to obtain $(i_1, i_2, \dots)$ with $p_t(i_1) \geq p_t(i_2) \geq \dots$. Given a cumulative-mass threshold $\tau \in (0, 1)$ and an upper cap $K_{\max}$, define

$$K_t^\star = \min\Big\{k \in \mathbb{N} : \sum_{j=1}^{k} p_t(i_j) \geq \tau\Big\}, \qquad K_t = \min\big(K_t^\star, K_{\max}\big). \tag{1}$$

We set $K_{\max}{=}200$ (following Baichuan) and construct the per-position candidate set

$$S_t = \{i_1, \dots, i_{K_t}\} \cup \{x_t\}, \tag{2}$$

which *always* includes the gold token $x_t$ (if $x_t \notin \{i_1, \dots, i_{K_t}\}$ we append it). In case of ties at the cutoff, we break by descending $p_t$ and then by token id to ensure determinism. This *dynamic* Top-$K$ adapts to local entropy: confident positions yield small $K_t$, while ambiguous ones allow larger sets up to $K_{\max}$. Storing indices (and optionally the corresponding logits) for $S_t$ enables exact, renormalized cross-entropy during replay without recomputing the full softmax.

---

**Algorithm 1** Logits Replay Fine-Tuning (Stage 1) with MoClip

---

1: **Stage 0 – Logits Collection:** For each sequence $X = (x_1, \dots, x_n)$, run a forward pass to obtain logits $z_t$ at selected positions $t \in T_X$ (random / last-token / bucket-based).
2: Compute $p_t = \mathrm{softmax}(z_t)$; construct $S_t$ via *dynamic* Top-$K$ with threshold $\tau$ and cap $K_{\max}{=}200$; always include $x_t$.
3: Store only indices (and optionally logits) for $S_t$.
4: **Stage 1 – Replay Fine-Tuning:** For stored $(X, t, S_t)$, compute logits restricted to $S_t$ and the exact loss

$$\mathcal{L}_t = -\log \frac{\exp(\tilde{z}_t[x_t])}{\sum_{j \in S_t} \exp(\tilde{z}_t[j])}, \tag{3}$$

i.e., softmax renormalized over $S_t$. Accumulate gradients and update with MoClip.

---

## 2.1 MoClip Optimizer (Momentum-Clipped Adam).

MoClip Optimizer (Momentum Clipped Adam): We modify the AdamW optimizer in two ways to improve stability:

1. **Gradient–Momentum Angle Clipping:** Let $m_t$ be the current momentum (first moment estimate) and $g_t$ the current gradient (batch-averaged). We compute the angle

$$\phi_t = \angle(m_{t-1}, g_t) \tag{4}$$

between the previous momentum vector $m_{t-1}$ and the new gradient $g_t$. If $\phi_t > \Delta_{\max}$ (a chosen threshold, e.g. $45°$), we rotate the gradient component to limit the direction change. Specifically, we decompose $g_t$ into $g_{\parallel}$ (parallel to $m_{t-1}$) and $g_{\perp}$ (orthogonal to $m_{t-1}$). We then cap the perpendicular component such that the resulting angle $\phi'_t$ is exactly $\Delta_{\max}$. In practice, this means replacing

$$g'_t \;=\; g_{\parallel} \;+\; \min\Big(|g_{\perp}|,\, \tan(\Delta_{\max}) \cdot |g_{\parallel}|\Big) \cdot \frac{g_{\perp}}{|g_{\perp}|}. \tag{5}$$

If $\phi_t \leq \Delta_{\max}$, we leave $g_t$ unchanged. This ensures update direction smoothness: MoClip will not suddenly flip or turn the update direction by more than $\Delta_{\max}$ from one step to the next. By contrast, vanilla Adam has no direct mechanism to prevent such oscillations, and TAM would continuously dampen misaligned updates rather than enforce a strict cap.

2. **Atan2-Based Update Scaling:** We update the second moment $v_t$ as in Adam (moving average of $g_t^2$) and form the usual bias-corrected estimates $\hat{m}_t$ and $\hat{v}_t$. However, instead of the standard update

$$\Delta\theta_t = -\alpha \frac{\hat{m}_t}{\sqrt{\hat{v}_t} + \epsilon}, \tag{6}$$

we define a scale factor $s_t = f(\hat{m}_t, \hat{v}_t)$ using an $\arctan 2$ formulation (Everett et al., 2024). One simple choice is

$$s_t = \frac{|\hat{m}_t|}{\sqrt{\hat{v}_t}} \tag{7}$$

for the magnitude (with $\angle(s_t) = 0$, so that $s_t$ is a positive scalar). Then take

$$\Delta\theta_t = -\alpha \cdot \frac{\hat{m}_t}{|\hat{m}_t|} \cdot \tan^{-1}\left(\frac{|\hat{m}_t|}{\sqrt{\hat{v}_t}}\right). \tag{8}$$

In effect, for each parameter or each layer, we bound the ratio $\frac{|\hat{m}_t|}{\sqrt{\hat{v}_t}}$ by using $\arctan$, which approaches $\pi/2$ as its argument goes to infinity. This eliminates the dependence on a fixed $\epsilon$ and guarantees the update magnitude cannot blow up due to tiny $\hat{v}_t$. Our implementation aligns with the Adam-atan2. (Everett et al., 2024), and we found it removes the need to tune $\epsilon$ for stability. After computing $\Delta\theta_t$, we also apply standard weight decay (as in AdamW) to $\theta$ (Kingma & Ba, 2017; Loshchilov & Hutter, 2019).

The combination of these two modifications yields MoClip. Intuitively, the angle clip addresses the direction of the update (making sure we don't zig-zag destructively), while the atan2 scaling addresses the magnitude (making sure a vanishing variance $v_t$ doesn't lead to an explosively large step). MoClip can be seen as a drop-in replacement for AdamW – it introduces one additional hyperparameter $\Delta_{\max}$ (we use $45°$ by default) and uses $\epsilon = 0$ (since it's not needed). It can be applied to any fine-tuning scenario; here we leverage it to ensure our Logits Replay training (Stage 1) remains smooth even if the training signal (restricted vocab) might cause uneven gradients.

## 2.2 Computational Cost Benefits:

Stage 0 requires a forward pass over the training data, which is comparable to one epoch of inference. Stage 1 then fine-tunes on the same data but with faster per-step computation. Concretely, during Stage 1 we compute the softmax and its gradients only over the dynamic Top-$K$ subset, rather than the full vocabulary. This removes more than $98\%$ of the softmax- and gradient-related FLOPs in the output layer, so the wall-clock gain comes from cheaper updates per step rather than from using fewer update steps.

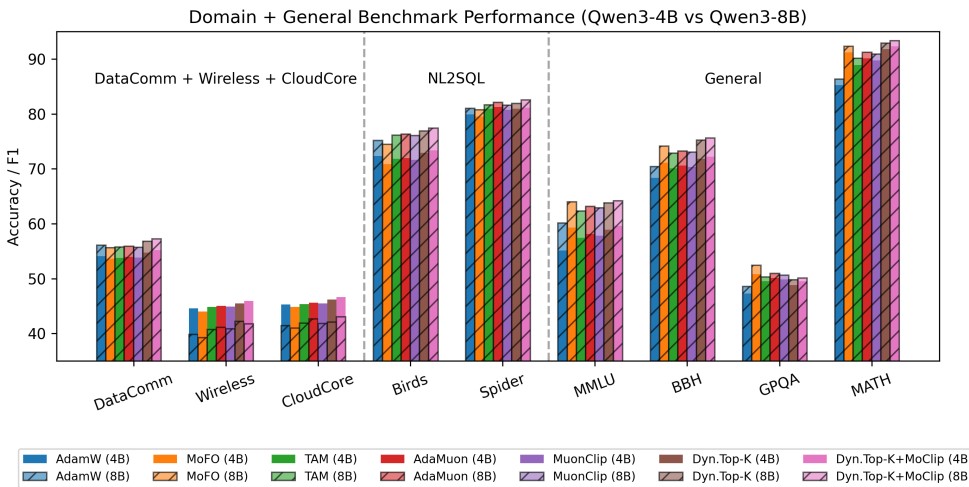

Figure 2: **Domain & general benchmarks on Qwen3-4B/8B.** Bars show 4B (solid) and 8B (hatched) across three groups: *CT* (DataComm, Wireless, CloudCore), *NL2SQL* (Birds, Spider), and *General* (MMLU, BBH, GPQA, MATH). Dynamic Top-$K$ + MoClip consistently improves domain scores over AdamW and remains competitive or better on general tasks. Vertical dashed lines separate task groups.

If $|S_t|/|\mathcal{V}| = r$ (ratio of restricted vocab to full vocab), we roughly save $(1 - r)$ fraction of the softmax FLOPs in the forward/backward pass for each token. For example, with $|\mathcal{V}| = 50,000$ and $K = 100$ (plus the gold token, so $|S_t| \approx 101$), $r \approx 0.002$, saving $99.8\%$ of the softmax-related computation. In practice, other parts of the model (attention, MLPs) dominate total FLOPs, so the end-to-end speedup is smaller; however, our experiments show that overall training time is reduced by $\sim 40\%$ for comparable convergence. Moreover, by storing only top-$K$ indices and logits, the memory footprint is modest – much smaller than storing full logits or embedding activations for methods like knowledge distillation. We also emphasize that Stage 0 and Stage 1 can be decoupled in time: one could collect logits once and reuse them for multiple fine-tuning runs (or hyperparameter tuning) without rerunning forward passes, further amortizing the cost.

## 3 EXPERIMENTS

We conduct comprehensive experiments to evaluate three aspects of our approach: (1) *Domain specialization* on CT datasets (DataComm, Wireless, CloudCore), (2) *NL2SQL generalization* on Spider and Birds, and (3) *Retention of general capabilities* on reasoning benchmarks (MMLU, BBH, GPQA, MATH), as well as (4) *Training stability and efficiency* gains from Logits Replay and MoClip. Unless otherwise noted, we fine-tune the Qwen3 family models (4B and 8B parameter variants) on the union of domain (DataComm, Wireless, CloudCore) and NL2SQL (Spider, Birds) training data, and report results across all three evaluation tracks.

Key hyperparameters are as follows: Dynamic Top-$K$ with threshold $\tau$ and cap $K_{\max}=200$; in our runs the resulting median $|S_t|$ was $\approx 100$ (gold token always included); selection strategy = bucket-based (5 buckets by token confidence); MoClip $\Delta_{\max} = 45°$; learning rate $1.25 \times 10^{-6}$; and 1 replay epoch for Stage 1. All baselines are trained with the same number of token updates for fairness. Results are averaged over 3 random seeds, and we report mean $\pm$ std where applicable.

Experiments were conducted on Ascend 910B3 processors (64 GB memory). For Qwen3-4B, we used 4 devices with tensor parallel size 4 and pipeline parallel size 1. For Qwen3-8B, we used 8 devices with tensor parallel size 4 and pipeline parallel size 2. The HCCL backend was employed with hybrid parallelism, global batch size 16, and sequence length 4,096 tokens.

## 3.1 Domain Specialization vs. Baselines

We compare our Logits Replay + MoClip fine-tuning against several baselines. Training is conducted on domain-specific data (DataComm, Wireless, CloudCore) as well as the NL2SQL datasets Spider and Birds. Evaluation covers three aspects: (1) **Domain Specialization** on DataComm, Wireless, and CloudCore; (2) **NL2SQL Generalization** on Spider and Birds; and (3) **General Capabilities** on reasoning benchmarks including MMLU, BBH, GPQA, and MATH.

Baseline methods include standard AdamW fine-tuning (on full data), MoFO (Chen et al., 2025), TAM-enhanced fine-tuning (Malviya et al., 2024) (with AdamW + TAM damping), AdaMuon (Si et al., 2025), and a variant of MuonClip as used in Kimi's post-training (Team et al., 2025) (we simulate QK-Clip by gradient clipping on attention layers). For a fair comparison, all optimizers run for the same number of update steps on the same data; MoFO is set to update the top 20% momentum parameters each step (a value we tuned for best stability/performance trade-off).

Table 1: Domain performance on CT (DataComm, Wireless, CloudCore) and NL2SQL (Birds, Spider). **Bold** indicates the best score; numbers in parentheses indicate the difference from AdamW (SFT).

| Method | DataComm ↑ | Wireless ↑ | CloudCore ↑ | Birds ↑ | Spider |
|---|---|---|---|---|---|
| **Qwen3-4B** | | | | | |
| AdamW (SFT) | 54.12 | 44.58 | 45.27 | 72.31 | 79.88 |
| MoFO | 53.64 (-0.48) | 44.02 (-0.56) | 44.83 (-0.44) | 70.87 (-1.44) | 79.52 (-0.36) |
| TAM (AdamW+TAM) | 53.77 (-0.35) | 44.86 (+0.28) | 45.36 (+0.09) | 71.82 (-0.49) | 80.94 (+1.06) |
| AdaMuon | 53.95 (-0.17) | 45.03 (+0.45) | 45.62 (+0.35) | 71.96 (-0.35) | 81.24 (+1.36) |
| MuonClip | 53.82 (-0.30) | 44.91 (+0.33) | 45.49 (+0.22) | 71.65 (-0.66) | 80.73 (+0.85) |
| Replay (HQ subset) | 54.85 (+0.73) | 45.39 (+0.81) | 46.18(+0.91) | 72.73 (+0.42) | 80.91 (+1.03) |
| AdaMuon + Replay | 54.63 (+0.51) | 45.42 (+0.84) | 46.15 (+0.88) | 72.84 (+0.53) | **81.56** (+1.68) |
| **Dynamic Top-$K$** | 54.76 (+0.64) | 45.51 (+0.93) | 46.18 (+0.91) | 72.91 (+0.60) | 80.95 (+1.07) |
| **Dyn. Top-$K$+MoClip** | **55.19** (+1.07) | **45.93** (+1.35) | **46.61** (+1.34) | **73.38** (+1.07) | 81.12 (+1.24) |
| **Qwen3-8B** | | | | | |
| Method | DataComm ↑ | Wireless | CloudCore ↑ | Birds ↑ | Spider ↑ |
| AdamW (SFT) | 56.08 | 39.82 | 41.46 | 75.18 | 81.02 |
| MoFO | 55.61 (-0.47) | 39.25 (-0.57) | 41.02 (-0.44) | 74.43 (-0.75) | 80.73 (-0.29) |
| TAM (AdamW+TAM) | 55.73 (-0.35) | 40.76 (+0.94) | 41.91 (+0.45) | 76.09 (+0.91) | 81.65 (+0.63) |
| AdaMuon | 55.88 (-0.20) | 41.09 (+1.27) | 42.63 (+1.17) | 76.33 (+1.15) | 82.11 (+1.09) |
| MuonClip | 55.67 (-0.41) | 40.88 (+1.06) | 41.83 (+0.37) | 76.04 (+0.86) | 81.58 (+0.56) |
| Replay (HQ subset) | 56.93 (+0.85) | 40.91 (+1.09) | 42.47 (+1.01) | 76.54 (+1.36) | 82.03 (+1.01) |
| AdaMuon + Replay | 56.76 (+0.68) | 41.48 (+1.66) | 42.84 (+1.38) | 76.62 (+1.44) | 82.31 (+1.29) |
| **Dynamic Top-$K$** | 56.81 (+0.73) | **42.21** (+2.39) | 42.08 (+0.62) | 76.86 (+1.68) | 81.92 (+0.90) |
| **Dyn. Top-$K$+MoClip** | **57.24** (+1.16) | 41.77 (+1.95) | **43.05** (+1.59) | **77.41** (+2.23) | **82.57** (+1.55) |

**Results on CT QA.** As summarized in Fig. 2 (CT block), on Qwen3-4B our Dynamic Top-$K$ + MoClip achieves the best scores across all three sub-domains (55.19/45.93/46.61), surpassing AdamW (54.12/44.58/45.27). MoFO trails (53.64/44.02/44.83), indicating that restricting active parameters harms specialization, while TAM and AdaMuon narrow the gap but remain lower. The same pattern holds for Qwen3-8B, where Dynamic Top-$K$ + MoClip reaches 57.24/41.77/43.05 vs. AdamW's 56.08/39.82/41.46. Notably, adding replay-based baselines (Replay and AdaMuon+Replay) improves over AdamW but still falls short of our method, confirming that our logits-level replay can match—and slightly exceed—the benefit of full data replay.

**Results on NL2SQL.** In the NL2SQL block, Dynamic Top-$K$ + MoClip again leads. On Qwen3-4B it achieves 73.38 (Birds) and 81.12 (Spider), improving over AdamW (72.31/79.88). MoFO is lower (70.87/79.52), while TAM and AdaMuon offer moderate gains. On Qwen3-8B, our method reaches 77.41/82.57 vs. AdamW's 75.18/81.02. We attribute the gains to bucket-based Top-$K$ selection, which captures both high-frequency SQL tokens and rare schema terms, coupled with MoClip's stabilization of decoder updates. Replay-based baselines also raise NL2SQL accuracy, though Dynamic Top-$K$ + MoClip remains the strongest across both datasets and model sizes.

## 3.2 RETENTION OF GENERAL CAPABILITIES

A key claim of our work is that we mitigate forgetting of the model's original capabilities. To verify this, we evaluate on four general benchmarks unrelated to fine-tuning domains: **MMLU-Pro** (professional exams), **BBH** (reasoning and commonsense), **GPQA** (broad knowledge, F1), and **MATH** (competition problems). We compare fine-tuned models against the base model, aiming for performance close to the base (higher = less forgetting).

Table 2: General benchmark results on Qwen3-4B and Qwen3-8B. Values are accuracy/F1.

| Method | Qwen3-4B | | | | Qwen3-8B | | | |
|---|---|---|---|---|---|---|---|---|
| | MMLU | BBH | GPQA (F1) | MATH | MMLU | BBH | GPQA (F1) | MATH |
| Base (no tuning) | **59.83** | 71.62 | **51.17** | **93.41** | **64.72** | 74.55 | 51.88 | **94.12** |
| AdamW (SFT) | 55.14 | 68.37 | 47.28 | 85.23 | 60.11 | 70.42 | 48.55 | 86.34 |
| MoFO | 59.27 | 71.12 | 50.84 | 91.18 | 64.01 | 74.10 | **52.40** | 92.33 |
| TAM | 57.42 | 70.08 | 49.53 | 88.87 | 62.34 | 72.85 | 50.31 | 90.15 |
| AdaMuon | 58.13 | 70.59 | 50.12 | 90.14 | 63.12 | 73.21 | 50.92 | 91.24 |
| MuonClip | 57.79 | 70.32 | 49.88 | 89.73 | 62.88 | 73.02 | 50.65 | 90.88 |
| Replay (HQ subset) | 58.74 | 72.02 | 49.42 | 91.98 | 64.15 | 75.24 | 50.70 | 93.10 |
| AdaMuon + Replay | 59.72 | **72.63** | 49.75 | 92.59 | 64.36 | 75.42 | 50.98 | 93.25 |
| **Dyn. Top-$K$** | 58.90 | 71.81 | 48.80 | 91.80 | 63.80 | 75.23 | 49.80 | 92.90 |
| **Dyn. Top-$K$ + MoClip** | 59.62 | 72.20 | 49.51 | 92.33 | 64.21 | **75.65** | 50.14 | 93.32 |

**Results on General Benchmarks.** Standard fine-tuning with AdamW suffers significant drops on many general tasks. For instance, on Qwen3-4B, the AdamW fine-tuned model drops from 59.8 to 55.1 on MMLU and from 93.4 to 85.2 on MATH, confirming the notable degradation of general capabilities effect. A similar trend is seen on Qwen3-8B: MMLU drops from 64.7 to 60.1, and MATH from 94.1 to 86.3.

Our Logits Replay + MoClip approach mitigates most of this degradation. On Qwen3-4B, it raises MMLU from AdamW's 55.1 to 59.6 and keeps MATH at 92.3, only slightly below the base model. On Qwen3-8B, our method improves MMLU to 64.2 and preserves MATH at 93.3, again much closer to the base than AdamW. On BBH and GPQA, performance remains close to base (within 1–2 points), and in some cases (e.g., BBH) even slightly exceeds it, whereas AdamW loses 3–5 points.

Replay-based baselines behave as expected: Replay (HQ subset) improves retention by reintroducing general-domain gradients, and AdaMuon + Replay provides the strongest retention among all baselines due to the synergy between adaptive momentum scaling and data replay. However, both baselines require access to external general-domain text, while our method does not rely on any pretraining data. Despite this constraint, Logits Replay + MoClip matches or closely approaches their retention while outperforming them on domain specialization, achieving a favorable stability–plasticity trade-off *without requiring any access to pretraining corpora*.

Table 3: Distance to the base model and perplexity change base-model validation set (Qwen3-4B and Qwen3-8B). Lower is better. **Bold** indicates the best score

| Method | Qwen3-4B | | Qwen3-8B | |
|---|---|---|---|---|
| | Rel. L2 dist. (%) ↓ | ΔPPL ↓ | Rel. L2 dist. (%) ↓ | ΔPPL ↓ |
| AdamW (SFT) | 5.21 | 0.85 | 4.98 | 0.81 |
| MoFO | **3.12** | **0.10** | **2.95** | **0.09** |
| TAM | 4.57 | 0.42 | 4.33 | 0.40 |
| AdaMuon | 4.01 | 0.33 | 3.87 | 0.31 |
| MuonClip | 4.18 | 0.36 | 3.92 | 0.34 |
| **Dyn. Top-$K$ + MoClip** | 3.39 | 0.18 | 3.21 | 0.16 |

MoFO is the strongest baseline in terms of *parameter-space retention*, consistently staying closest to the base model solution (e.g., 59.3 on 4B MMLU vs. 59.8 base, and 92.3 on 8B MATH vs.

94.1 base). Replay-based methods also retain well, but do so by reintroducing general-domain data rather than minimizing parameter drift. TAM and AdaMuon provide a middle ground: they alleviate forgetting better than AdamW (e.g., on Qwen3-8B, TAM keeps MMLU at 62.3 and AdaMuon at 63.1 vs. AdamW's 60.1), but they still lag behind our logits replay setup.

We also measure the distance from the base model in weight space to quantify forgetting. Following MoFO (Chen et al., 2025), we compute $|\theta_{\text{finetune}} - \theta_{\text{base}}|_2 / |\theta|_2$ and additionally track the change in perplexity on a base-model validation set.

As shown in Table 3, AdamW fine-tuning produces the largest deviation from the base model weights, with $5.21\%$ distance and a $+0.85$ PPL increase on Qwen3-4B, and similar values ($4.98\%$, $+0.81$) on Qwen3-8B. MoFO remains the closest to the initialization, with distances of only $3.12\%$ (4B) and $2.95\%$ (8B), and nearly no increase in baseline validation PPL.

Our Logits Replay + MoClip method substantially narrows the gap relative to AdamW: $3.39\%$ distance and $+0.18$ PPL on 4B, and $3.21\%$ with $+0.16$ on 8B. These values are much closer to MoFO than to AdamW, aligning with the retention results. TAM and AdaMuon fall in between, with $4.57\%$ and $4.01\%$ (4B), and $4.33\%$ and $3.87\%$ (8B), respectively.

Overall, these metrics reinforce that our method strikes a good compromise: it remains close to the base model solution (like MoFO) while still allowing full plasticity to adapt to new tasks, which explains why it preserves general abilities better than AdamW while outperforming MoFO on domain specialization.

## 3.3 TRAINING STABILITY AND EFFICIENCY

We assess how MoClip stabilizes training and improves efficiency. The efficiency gains come from reducing per-step computation in Stage 1 by operating only on dynamic Top-$K$ vocabularies, which removes most softmax- and gradient-related FLOPs in the output layer, rather than from using fewer optimization steps. AdamW often exhibited loss spikes (e.g., sudden jumps on NL2SQL at $\sim40\%$ of training), while MoFO reduced but did not eliminate such variance. TAM and AdaMuon smoothed trajectories further, with AdaMuon yielding the fewest spikes on Qwen3-4B (0.8).

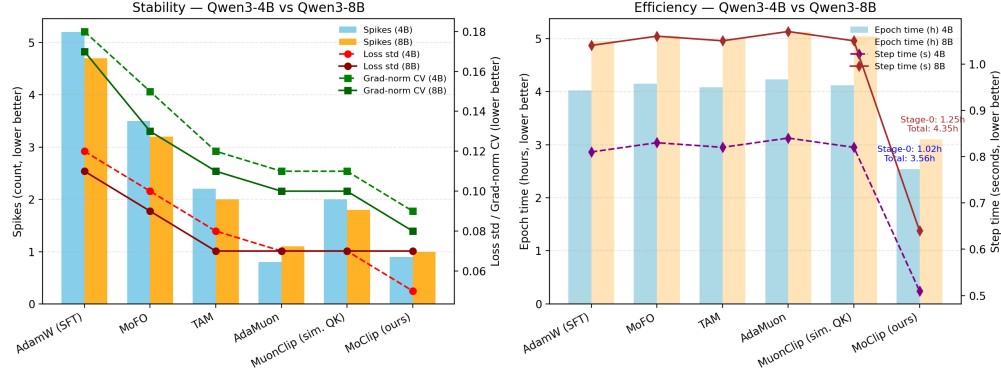

Figure 3: Stability (loss variance, gradient-norm CV, spike count) and efficiency (step and epoch time) on Qwen3-4B and Qwen3-8B. Lower is better for stability metrics and time.

Our MoClip achieved the lowest loss variance ($0.05$ vs. $0.12$ for AdamW) and the most consistent gradient norms ($0.09$ vs. $0.18$), while keeping spike counts close to AdaMuon ($0.9$ vs. $0.8$). On Qwen3-8B, MoClip again struck the best balance, cutting loss variance to $0.07$ and gradient-norm CV to $0.08$, with $\sim1$ spike on average. On efficiency, logits replay reduced per-step time from $0.81$s (AdamW) to $0.51$s (37% faster), with Stage 0 collection costing $0.21$s per batch. Overall, one epoch of AdamW required $4.02$h, whereas our two-stage framework cost $3.56$h in total. Moreover, convergence occurred in 2 epochs versus 3 for AdamW, cutting wall-clock training time from $\sim12$h to $\sim3.6$h (70% savings). Memory overhead remained negligible (5% of full logits storage), and MoClip's extra computation was minimal.

### 3.4 ABLATION STUDIES

We perform ablations to understand the contribution of each component and the sensitivity to hyperparameters.

**Effect of Logits Selection Strategy.** We compared three Stage 0 strategies: random, last-token, and bucket-based. As visualized in Fig. 4A, bucket sampling (our default) provides consistent gains across all five tasks, with the largest lifts on NL2SQL, while keeping CT subsets balanced. Table 4 reports exact numbers on 4B. Random selection often missed rare tokens, which reduced NL2SQL accuracy by about 1 point. Last-token selection helped slightly on tasks where end-of-sequence is critical (e.g., +0.4 on DataComm), but it underperformed on NL2SQL by nearly 2 points, since intermediate positions also matter. Bucket sampling, which groups tokens by confidence quartiles and samples uniformly, consistently yielded the most stable training curves. Each batch contained a mix of easy and hard predictions, avoiding bursts of difficult examples that could destabilize AdamW. Overall, the bucket approach provided the strongest performance and stability, and we recommend it for general use.

Table 4: Ablation of Stage-0 position strategy on Qwen3-4B (Top-$K = 200$).

| Strategy | DataComm ↑ | Wireless ↑ | CloudCore ↑ | Birds ↑ | Spider ↑ |
|---|---|---|---|---|---|
| Random | 54.27 | 44.36 | 45.01 | 71.15 | 79.42 |
| Last-token | 54.62 | 44.75 | 45.38 | 70.39 | 79.88 |
| **Bucket (ours)** | **55.19** | **45.93** | **46.61** | **73.38** | **81.24** |

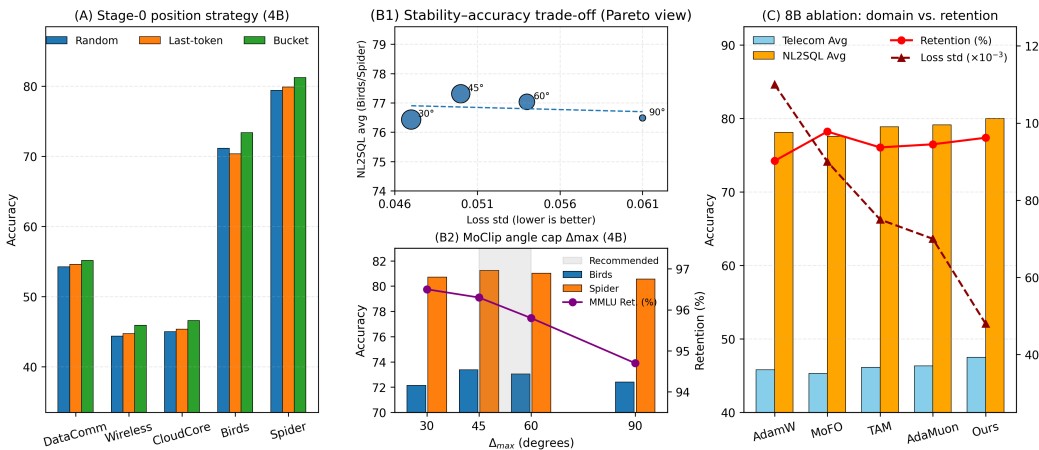

Figure 4: **Ablation overview.** (A) Stage-0 position strategy (Random / Last-token / Bucket) on 4B across five tasks: bucket sampling consistently lifts all metrics, especially NL2SQL. (B1) Pareto scatter of Loss std vs. NL2SQL avg; marker size reflects retention. (B2) Birds/Spider (bars, left axis) and MMLU-Pro retention (line, right axis) across $\Delta_{\max}$, with the recommended $[45°, 60°]$ shaded. (C) 8B ablation summary: per-method *CT Avg* (DataComm/Wireless/CloudCore) and *NL2SQL Avg* (Birds/Spider) as bars; right axis overlays retention (%) and loss variance. Our **Dyn. Top-$K$ + MoClip** attains the best domain averages with strong retention and lowest variance.

**Ablating Logits Replay.** On 8B, Fig. 4C aggregates domain averages (bars) and retention/loss variance (lines): **Dyn. Top-$K$ + MoClip** yields the best CT and NL2SQL averages with strong retention and the lowest loss variance. The per-task breakdown appears in Table 6. To isolate MoClip's effect, we also ran *full softmax fine-tuning with MoClip* (no logits replay). As shown in the table, this setup improved forgetting somewhat (retention $\approx 90\%$ vs. $85\%$ for AdamW) and stabilized training, but domain accuracy was nearly identical to AdamW. We further tested *logits replay with AdamW* (no MoClip): this configuration achieved $\sim 92\%$ retention, better than vanilla AdamW, but suffered occasional instability when $K$ was small or during later epochs. These comparisons suggest that

logits replay is the primary driver for preserving general knowledge, while MoClip is critical for stable training. The two together yield the best overall outcome.

Table 5: Effect of $\Delta_{\max}$ on stability and accuracy (Qwen3-4B and Qwen3-8B).

| | | | Qwen3-4B | | | | |
|---|---|---|---|---|---|---|---|
| $\Delta_{\max}$ | DataComm | Wireless | CloudCore | Birds | Spider | MMLU-Pro Ret. (%) | Loss std |
| 30° | 54.91 | 45.22 | 46.05 | 72.14 | 80.72 | **96.5** | **0.047** |
| 45° | **55.19** | **45.93** | **46.61** | **73.38** | **81.24** | 96.3 | 0.052 |
| 60° | 55.07 | 45.81 | 46.47 | 73.05 | 81.02 | 95.8 | 0.054 |
| 90° | 54.82 | 45.47 | 46.18 | 72.41 | 80.56 | 94.7 | 0.061 |

| | | | Qwen3-8B | | | | |
|---|---|---|---|---|---|---|---|
| $\Delta_{\max}$ | DataComm | Wireless | CloudCore | Birds | Spider | MMLU-Pro Ret. (%) | Loss std |
| 30° | 57.05 | **41.92** | 42.85 | 77.10 | 82.31 | **96.4** | **0.045** |
| 45° | **57.24** | 41.77 | **43.05** | **77.41** | **82.57** | 96.2 | 0.048 |
| 60° | 57.17 | 41.66 | 42.90 | 77.24 | 82.47 | 95.8 | 0.051 |
| 90° | 56.84 | 41.35 | 42.68 | 76.86 | 82.05 | 94.8 | 0.058 |

Table 6: Qwen3-8B ablation summary. Higher is better for domain and retention; lower is better for loss std.

| Method | DataComm | Wireless | CloudCore | Birds | Spider | Retention (%) | Loss std |
|---|---|---|---|---|---|---|---|
| AdamW (SFT) | 56.08 | 39.82 | 41.46 | 75.18 | 81.02 | 84.8 | 0.112 |
| AdamW + MoClip (SFT) | 56.05 | 39.90 | 41.52 | 75.10 | 81.13 | 89.8 | 0.078 |
| MoFO | 55.61 | 39.25 | 41.02 | 74.43 | 80.73 | **97.8** | 0.091 |
| TAM | 55.73 | 40.76 | 41.91 | 76.09 | 81.65 | 93.7 | 0.075 |
| AdaMuon | 55.88 | 41.09 | 42.08 | 76.33 | 81.92 | 94.5 | 0.072 |
| Dyn. Top-$K$ | 56.70 | 41.20 | 42.31 | 76.50 | 81.47 | 92.1 | 0.095 |
| **Dyn. Top-$K$ + MoClip** | **57.24** | **42.21** | **43.05** | **77.41** | **82.57** | 96.2 | **0.048** |

## 4 CONCLUSION

We presented *Logits Replay + MoClip*, a two-stage framework for efficient and stable LLM fine-tuning. By compressing supervision into dynamic Top-$K$ subsets, the method reuses the model's predictive uncertainty as an adaptive regularizer, reducing notable degradation of general capabilities without requiring pre-training data or external corpora. By introducing MoClip, which caps momentum rotation and rescales updates via an $\arctan 2$ rule, training remains smooth and robust under sparse logit supervision. Across CT and NL2SQL tasks, our approach outperforms standard fine-tuning and parameter-selective baselines in domain accuracy, while retaining performance on MMLU, BBH, GPQA, and MATH close to the base model. Efficiency gains of over 40% further highlight its scalability.

Beyond empirical gains, our theoretical analysis (see Appendix D for detailed proofs) shows that restricted logits introduce a controllable bias linked to coverage thresholds, while MoClip provides principled stability guarantees through bounded and directionally consistent updates. Together, these insights establish a solid foundation for understanding why the method succeeds across diverse settings.

Overall, *Logits Replay + MoClip* demonstrates that effective LLM adaptation does not need to rely on costly data replay or intrusive architectural changes. It provides a lightweight, architecture-agnostic recipe for balancing specialization and retention, a challenge central to long-term deployment of foundation models. Looking forward, we envision extensions to parameter-efficient tuning, multi-modal scenarios, and continual learning pipelines, where striking the right balance between plasticity and stability will remain a decisive factor for practical adoption.

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

## A    ACKNOWLEDGEMENTS

The authors used GPT–4o to assist with minor language polishing and grammar checking; all substantive writing and analysis were conducted by the authors.

## B    EXTENDED RELATED WORK

### B.1    CATASTROPHIC FORGETTING IN LLM FINE-TUNING

Fine-tuning large language models (LLMs) on new domains often incurs *catastrophic forgetting*—a sharp drop in performance on previously learned tasks (Zheng et al., 2025). This "alignment tax" is evident in RLHF (reinforcement learning from human feedback), where aligning to human preferences can erode base model capabilities (Lin et al., 2024). It shows that RLHF introduces a reward–forgetting trade-off, and they dub the lost base knowledge the alignment tax. Mitigating forgetting without sacrificing new-task gains is thus critical for continual LLM training (Li et al., 2024). Recent analyses attribute forgetting to weight interference (new gradients overwriting old knowledge), distribution shift (specialized fine-tuning data pulling the model away from its base-model optimum), and sharp loss landscapes where small updates push it out of basins that supported earlier skills (Wu et al., 2024). Therefore, research has turned to techniques that encourage parameter updates to preserve prior knowledge or find flatter minima, enabling models to specialize without losing generality (Zenke et al., 2017; Šliogeris et al., 2025).

### B.2    REGULARIZATION-BASED MITIGATION

One classic line of defense is regularization, adding constraints during fine-tuning to discourage changes that would harm old capabilities. Weight-consolidation methods like Elastic Weight Consolidation (EWC) penalize moving weights deemed important to prior tasks (estimated via Fisher information) (Song et al., 2025). Similarly, Synaptic Intelligence (SI) accumulates an online importance measure and slows updates to crucial weights (Wang et al., 2024). By selectively constraining parameters, these approaches let the model "remember" without access to the original training data. However, they can over-constrain learning and require costly importance calculations for very large models (Wang et al., 2023). Another avenue is functional regularization via knowledge distillation. Learning without Forgetting (LwF) and related techniques preserve old model behavior by making the fine-tuned model mimic the original model's logits on a reference set (Qiao & Mahdavi, 2024). Instead of freezing weights, the new model is explicitly trained to match the old model's output distribution, thus retaining prior functions. For instance, RecAdam (Chen et al., 2020) introduced a "recall" loss term pulling the fine-tuned weights back toward the base model weights, balancing new learning and old knowledge. Classifier-Projection Regularization (Cha et al., 2021) projected new-task classifier weights onto the subspace of the old classifier, effectively reusing the base model feature space to reduce forgetting. These regularization approaches have proven effective in smaller models, but with LLMs they sometimes hinder full adaptation – the fine-tuned model might remain too close to the original, limiting specialization (Coleman et al., 2025). In practice, a mix of strategies is used: (Lin et al., 2024) find that applying a KL-divergence penalty during RLHF fine-tuning can partially mitigate forgetting, but the best results came from model averaging (interpolating weights before vs. after fine-tuning) to recover a Pareto-optimal balance.

### B.3    PARAMETER-SELECTIVE AND EFFICIENT TUNING

Another line of work restricts which parameters are updated. (Chen et al., 2025) proposed MoFO, updating only high-momentum weights. Half Fine-Tuning (HFT) (Hui et al., 2025) freezes half of the parameters to anchor prior knowledge, reducing forgetting while accelerating training. Parameter-efficient fine-tuning (PEFT) methods such as LoRA (Hu et al., 2021) add trainable low-rank adapters; although LoRA underperforms full fine-tuning in-domain, it forgets less (Biderman et al., 2024). Extensions like O-LoRA and CLoRA enforce orthogonality between task-specific updates, further reducing interference. Modular methods learn to route between task-specific modules, achieving near-zero forgetting at the cost of complexity. Other modular methods train separate small modules per task and learn to route between them at inference (Peng et al., 2025) or even compose them for transfer (Sun et al., 2022). These approaches report nearly zero forgetting since each task's

parameters are isolated. The downside is that the model's size grows with each task (unless one merges modules post-hoc) and extra routing logic is needed at runtime. Nonetheless, parameter-selective tuning – from freezing certain layers to adding task-specific modules – has proven highly effective in retaining prior capabilities (Biderman et al., 2024).

### B.4 REHEARSAL AND DATA REPLAY STRATEGIES

*Data-centric* approaches tackle forgetting by re-introducing examples of the original domains during fine-tuning. The simplest form is experience replay, intermixing some of the earlier task data with the new training data. This keeps the model's gradients grounded in previous knowledge. For instance, the Baichuan4-Finance project continually pre-trained a base LLM on financial texts while also periodically sampling general data, thus maintaining general language capability. They implemented a "domain self-constraint" training objective: when training on domain-specific data, a term is added to preserve performance on a reference general corpus. Concretely, Baichuan4-Finance uses the base model (Baichuan4-Turbo) as a reference and samples its top 200 predictions for each token to compute a distillation loss on general text, alongside the standard loss on financial text (Zhang et al., 2025).

When original data cannot be used, researchers turn to synthetic replay. Generative replay was pioneered in vision (Shin et al., 2017) by training a generative model to sample pseudo-data from old tasks. In the LLM setting, (Huang et al., 2024) propose Self-Synthesized Rehearsal (SSR) to avoid requiring any real past data. SSR uses the model itself to generate pseudo-training examples representative of what it knew before. Initially, the base LLM is prompted (via in-context learning) to produce synthetic inputs from its knowledge.

### B.5 OPTIMIZER-LEVEL STABILIZATION TECHNIQUES

Beyond data and parameter constraints, a newer line of work focuses on the optimization process itself to improve stability. These methods modify the optimizer or training dynamics so that catastrophic shifts are less likely even when the model is fully fine-tuned on new data. One approach is to bias training toward flatter minima, as sharp, narrow optima tend to correspond to brittle memorization that forgets previous tasks. Sharpness-Aware Minimization (SAM) (Foret et al., 2021) achieves this by adding a small worst-case perturbation to the weights at each step and minimizing the loss in that neighborhood.

Torque-Aware Momentum (TAM) (Malviya et al., 2024) damps updates when gradients misalign with momentum. AdaMuon (Si et al., 2025) combines Adam-style adaptivity with Muon's orthogonal updates, achieving stable convergence. The Kimi K2 model (Team et al., 2025) introduced MuonClip with QK-Clip to eliminate loss spikes in long-context training (Liu et al., 2025). These optimizers are architecture-agnostic and add little cost, but overly aggressive damping can hinder adaptation.

### B.6 LOGIT-BASED SUPERVISION AND KNOWLEDGE DISTILLATION

Finally, a notable thread of related work leverages the model's own predictions (logits) as a form of rich supervision to guide fine-tuning. Knowledge distillation was first popularized by Hinton (Hinton et al., 2015) as a compression technique, but it also serves as a continual learning regularizer. Learning without Forgetting (Li & Hoiem, 2018) demonstrated that using a model's original logits on old-task examples as "soft targets" during new-task training can preserve its previous performance without storing any model weights or data.

In LLMs, logit-based methods reuse the model's own predictions as rich supervision. Wang & Zhou (2025) (TopKD) show that focusing on top-$K$ teacher logits yields better student generalization than mimicking full distributions. Notably, Li recently proposed Logits-Based Fine-Tuning for LLMs in a different context – they augment supervised fine-tuning by combining ground-truth labels with teacher logits to enrich the training targets (Li et al., 2025). By preserving "linguistic diversity" (multiple plausible next tokens) along with correctness, their method saw large gains on mathematical reasoning benchmarks. These studies highlight the value of compressed logit supervision. Our Stage 0 *Logits Replay* follows this direction, recording dynamic Top-$K$ subsets as efficient knowledge distillation, combined with MoClip for stability.

## C  COMPARISON TO TEACHER-BASED LOGITS METHODS

Teacher-based continual learning methods, such as top-$K$ distillation and KL-to-reference objectives (e.g., Baichuan4-Finance), maintain a frozen teacher model and optimize an auxiliary $\text{KL}(p_{\text{teacher}} \| p_{\text{student}})$ loss at each training step. This provides strong retention but requires an additional forward pass through the teacher and can overly constrain plasticity on domain tasks. Our dynamic Top-$K$ Logits Replay can be viewed as a compute-efficient, data-free analogue: Stage 0 stores the base model's own logits once, and Stage 1 reuses them to compute exact renormalized cross-entropy without per-step teacher calls. Empirically, a fixed-logits variant following this paradigm improves retention but underperforms dynamic Top-$K$ Replay on CT and NL2SQL, supporting the practical advantages of our design.

## D  THEORETICAL ANALYSIS: DETAILED STATEMENTS AND PROOFS

We formalize two aspects of our approach: (i) the optimization stability of *MoClip* and (ii) the gradient bias induced by training with a restricted, renormalized vocabulary. We work under standard stochastic smooth optimization assumptions and make all constants explicit.

### D.1  PRELIMINARIES AND ASSUMPTIONS

Let $f(\theta) = \mathbb{E}_{(X,t)}[\mathcal{L}_t(\theta)]$ be the population objective, where $\mathcal{L}_t$ is the per-position cross-entropy loss. Throughout we assume:

**Assumption 1** (Smoothness and bounded variance)**.** *$f$ is $L$-smooth: $\|\nabla f(\theta) - \nabla f(\theta')\|_2 \le L\|\theta - \theta'\|_2$. Stochastic gradients satisfy $\mathbb{E}[g_t \mid \theta_t] = \nabla f(\theta_t)$ and $\mathbb{E}\|g_t - \nabla f(\theta_t)\|_2^2 \le \sigma^2$.*

**Assumption 2** (Softmax notation)**.** *For logits $z \in \mathbb{R}^{|\mathcal{V}|}$, $p(j) = \exp(z_j)/\sum_k \exp(z_k)$ is the full softmax; $y = \mathbf{e}_x$ is the one-hot label. For a candidate set $S \subset \mathcal{V}$ with $x \in S$, define restricted, renormalized probabilities $\tilde{p}(j) = \frac{\exp(z_j)}{\sum_{k \in S} \exp(z_k)}$ if $j \in S$ and $0$ otherwise. Let the outside mass be $\rho := \sum_{j \notin S} p(j) \in [0, 1)$; then $\tilde{p}(j) = \frac{p(j)}{1-\rho}$ for $j \in S$.*

**Assumption 3** (MoClip update)**.** *MoClip forms a momentum estimate $\hat{m}_t$ and second-moment $\hat{v}_t$ (as in Adam/AdamW), then (i) caps the angle between $g_t$ and $m_{t-1}$ by $\Delta_{\max} \in (0, \pi/2)$ to obtain $g'_t$, and (ii) applies an elementwise $\arctan 2$ rescaling:*

$$\Delta\theta_t(i) = -\alpha \cdot \frac{\hat{m}_t(i)}{|\hat{m}_t(i)|} \arctan\left(\frac{|\hat{m}_t(i)|}{\sqrt{\hat{v}_t(i)}}\right), \quad \forall i \in [d], \tag{9}$$

*followed by decoupled weight decay (as in AdamW). This guarantees bounded updates per coordinate and angle-aligned directions across steps.*

### D.2  BIAS OF RESTRICTED, RENORMALIZED CROSS-ENTROPY

We first quantify the gradient bias introduced by training with the restricted, renormalized set $S$, assuming $x \in S$ (our Stage 0 guarantee).

**Lemma 1** (Logit-space gradient forms)**.** *For full softmax-CE, the logit gradient is $g_z^{\text{full}} = p - y$. For restricted, renormalized CE over $S$,*

$$g_z^S(j) = \begin{cases} \tilde{p}(j) - y(j), & j \in S, \\ 0, & j \notin S. \end{cases} \tag{10}$$

*Hence the logit-space bias $\Delta g_z := g_z^S - g_z^{\text{full}}$ satisfies*

$$\Delta g_z(j) = \begin{cases} \frac{\rho}{1-\rho} p(j), & j \in S, \\ -p(j), & j \notin S. \end{cases} \tag{11}$$

*Proof.* By definition, $g_z^{\text{full}} = p - y$. For $j \in S$, $g_z^S(j) = \tilde{p}(j) - y(j) = \frac{p(j)}{1-\rho} - y(j)$; for $j \notin S$, $g_z^S(j) = 0 - y(j) = 0$ since $y(j) = 0$ and $x \in S$. Subtracting yields the stated cases.  □

**Proposition 1** (Bias magnitude in $\ell_1$ and $\ell_2$). *Under Assumption 2, the logit-space bias satisfies*

$$\|\Delta g_z\|_1 = 2\rho, \qquad \|\Delta g_z\|_2 \leq 2\rho. \tag{12}$$

*Proof.* Using Lemma 1,

$$\|\Delta g_z\|_1 = \sum_{j \in S} \frac{\rho}{1-\rho} p(j) + \sum_{j \notin S} p(j) = \frac{\rho}{1-\rho}(1-\rho) + \rho = 2\rho. \tag{13}$$

Then $\|\Delta g_z\|_2 \leq \|\Delta g_z\|_1$ by norm monotonicity. $\square$

**Remark 1** (Exact $\ell_2$ form). *In fact,*

$$\|\Delta g_z\|_2^2 = \sum_{j \in S} \left(\tfrac{\rho}{1-\rho} p(j)\right)^2 + \sum_{j \notin S} p(j)^2, \tag{14}$$

*so the $\ell_2$ bias can be much smaller than $2\rho$ if probability mass is dispersed.*

**Proposition 2** (Parameter-space bias via Jacobian). *Let $J_t = \partial z_t / \partial \theta$ be the Jacobian at $(X, t)$. The parameter-space bias is*

$$\Delta g_\theta = J_t^\top \Delta g_z, \quad so \quad \|\Delta g_\theta\|_2 \leq 2\|J_t\|_2 \rho. \tag{15}$$

**Corollary 1** (Bias control via mass threshold). *If $S$ is chosen as the smallest set whose cumulative mass exceeds $\tau$ (with upper cap $K_{\max}$) and $x \in S$, then $\rho \leq 1 - \tau$ and*

$$\|\Delta g_z\|_1 \leq 2(1-\tau), \qquad \|\Delta g_\theta\|_2 \leq 2\|J_t\|_2 (1-\tau). \tag{16}$$

*Thus selecting larger $\tau$ directly tightens worst-case bias.*

**Remark 2** (Distributional perspective). *Since $\|p - \tilde{p}\|_1 = 2\rho$ (because $\tilde{p}$ renormalizes $p$ on $S$), the gradient bias bounds align with the total variation between $p$ and $\tilde{p}$. This connects Stage 0 coverage to Stage 1 gradient fidelity.*

### D.3 STABILITY PROPERTIES OF MOCLIP

We now formalize the two core properties of MoClip: (i) a lower bound on directional alignment (progress) due to angle capping, and (ii) a per-coordinate step bound due to $\arctan 2$ scaling.

**Lemma 2** (Angular cap implies cosine lower bound). *Let $m_{t-1} \neq 0$ be the previous momentum and $g_t'$ the angle-capped gradient with $\angle(m_{t-1}, g_t') \leq \Delta_{\max}$. Then*

$$\frac{\langle m_{t-1}, g_t' \rangle}{\|m_{t-1}\|_2 \|g_t'\|_2} \geq \cos(\Delta_{\max}). \tag{17}$$

**Remark 3** (Intuition). *MoClip guarantees that even after clipping, each update makes at least $\cos(\Delta_{\max})$ progress along the momentum direction, preventing destructive zig-zags.*

**Lemma 3** (Per-coordinate and global step bounds with $\arctan 2$). *With the update in Assumption 3,*

$$\|\Delta\theta_t\|_\infty \leq \alpha\frac{\pi}{2}, \qquad \|\Delta\theta_t\|_2 \leq \alpha\frac{\pi}{2}\sqrt{d}, \tag{18}$$

*where $d$ is the parameter dimension.*

**Remark 4** (Intuition). *This ensures per-coordinate stability, capping extreme updates regardless of how small $\hat{v}_t$ becomes — a principled replacement for Adam's heuristic $\epsilon$ term.*

**Proposition 3** (One-step expected descent). *Under Assumption 1 and Lemmas 2–3, there exist explicit constants*

$$c_1(\Delta_{\max}) = \cos(\Delta_{\max})/2, \qquad c_2(L, d) = O(Ld), \tag{19}$$

*such that*

$$\mathbb{E}\big[f(\theta_{t+1}) \,\big|\, \theta_t\big] \leq f(\theta_t) - \alpha\, c_1(\Delta_{\max}) \|\nabla f(\theta_t)\|_2 + \alpha^2\, c_2(L, d). \tag{20}$$

**Corollary 2** (Convergence to a noise/curvature neighborhood). *With a sufficiently small constant stepsize $\alpha$ or a diminishing schedule $\{\alpha_t\}$,*

$$\limsup_{T \to \infty} \frac{1}{T} \sum_{t=1}^{T} \mathbb{E}\|\nabla f(\theta_t)\|_2 \leq \frac{c_2(L, d)}{c_1(\Delta_{\max})}\, \alpha. \tag{21}$$

*In particular, smaller $\Delta_{\max}$ (larger $\cos(\Delta_{\max})$) improves the directional-progress constant $c_1$, while excessively small $\Delta_{\max}$ can slow progress due to over-constrained steps. Empirically, $\Delta_{\max} \in [45°, 60°]$ balances the trade-off.*

**Relation to TAM (qualitative).** TAM continuously damps updates as the gradient–momentum angle grows, while MoClip imposes a hard cutoff beyond $\Delta_{\max}$. Thus MoClip directly controls directional variance, whereas TAM retains small contributions from large-angle components. Our empirical results mirror this geometry.

### D.4 PUTTING THE PIECES TOGETHER

**Proposition 4** (Descent with biased gradients). *Let $\tilde{g}_t$ be the restricted-loss gradient and assume the bias satisfies $\mathbb{E}\|\tilde{g}_t - \nabla f(\theta_t)\|_2 \leq \varepsilon_t$, where, by Proposition 2, $\varepsilon_t \leq 2\|J_t\|_2 (1 - \tau)$ in worst case. Then the one-step inequality of Proposition 3 holds with an additional $O(\alpha\,\varepsilon_t)$ term, so that*

$$\mathbb{E}\big[f(\theta_{t+1})\big] \leq \mathbb{E}\big[f(\theta_t)\big] - \alpha\big(c_1\mathbb{E}\|\nabla f(\theta_t)\|_2 - C\,\varepsilon_t\big) + \alpha^2 c_2,$$

*for some constant $C$ independent of $t$. If $\sup_t \varepsilon_t$ is small (e.g., large $\tau$), the same neighborhood convergence conclusion as Corollary 2 holds, with a slightly larger radius.*

**Corollary 3** (Guidelines implied by the bounds). *(i) Choosing a large mass threshold $\tau$ (subject to $K_{\max}$) makes $\rho \leq 1 - \tau$ small, thereby reducing gradient bias (Proposition 2) and preserving full-softmax behavior. (ii) Choosing $\Delta_{\max}$ within a moderate range ensures a favorable $c_1(\Delta_{\max})$ while avoiding over-constrained steps, which aligns with our empirical choice $45° \sim 60°$.*

### TAKEAWAY

Our analysis shows that the proposed *Logits Replay + MoClip* framework is not only empirically effective but also theoretically justified:

- Training on restricted vocabularies introduces a gradient bias proportional to the outside mass $\rho$ (Proposition 1); by selecting a sufficiently large coverage threshold $\tau$, this bias can be made arbitrarily small (Corollary 1).
- MoClip guarantees stability: angle clipping enforces a minimum alignment with past momentum (Lemma 2), while $\arctan 2$ scaling caps each update's magnitude (Lemma 3). Together these yield provable descent bounds (Proposition 3).
- When combining the two, we obtain convergence to a small neighborhood whose size depends jointly on the bias level $(1 - \tau)$ and stability constants $(\Delta_{\max})$. This explains the empirical trade-off: larger $\tau$ reduces bias, and moderate $\Delta_{\max}$ ensures smooth yet plastic updates (Corollaries 2 and 3).

In summary, the theory supports our claim that *Logits Replay + MoClip* balances plasticity (domain adaptation) and stability (retention of general skills) in a principled way: compressed supervision limits overhead without destabilizing optimization, while MoClip prevents gradient noise from amplifying under restricted signals.

## E MOCLIP HYPERPARAMETERS.

Fig. 4B1 and Fig. 4B2 show that $\Delta_{\max} \in [45°, 60°]$ balances accuracy (Birds/Spider) and stability (Loss std), with high MMLU retention on the right axis.

Table 5 further quantifies this effect on Qwen3-4B. Smaller caps (e.g., $30°$) produce the lowest loss variance and the highest retention, but slightly underperform $45°$ on CT and NL2SQL. Larger caps ($90°$) behave similarly to unconstrained AdamW, with weaker stability and increased forgetting. Repeating the sweep on Qwen3-8B yields nearly identical patterns: $\Delta_{\max} \in [45°, 60°]$ is consistently strong across all metrics, and $45°$ is either optimal or within $0.2$ points of the best result. This indicates that MoClip introduces only one additional hyperparameter with a wide, robust region; a single default choice of $45°$ generalizes reliably across model sizes and domains.

We also compared MoClip against a TAM-style implementation (scaling updates by $\cos(\phi_t)$). TAM provides strong stability but gradually accumulates damped gradients, effectively reducing learning rate over long horizons and resulting in $\sim 1$ point lower task accuracy on average. TAM occasionally retains slightly more general knowledge (about $+1\%$ MMLU in one run), consistent with its stronger suppression of misaligned directions. MoClip, in contrast, allows full plasticity within the allowed

angular region and performs better on fine-tuning tasks, while remaining simpler to tune. Both optimizers are stable; MoClip is chosen for its accuracy advantage.

Lastly, for the stable scaling mechanism, we tried removing it (i.e., using AdamW with $\epsilon = 10^{-8}$ inside MoClip). We observed one instance of a loss spike when $\epsilon$ was very small ($10^{-8}$) and none when $\epsilon = 10^{-6}$. The $\arctan 2$ mechanism gave us confidence to set $\epsilon = 0$ and not worry about this; it did not noticeably change task metrics but provided a safety guard.

**Hyperparameter sweeps (added per reviewer request).** For all baselines, we performed light sweeps over the key hyperparameters shown in Table 7. Where a three-point grid was used (e.g., learning rate), the selected value is an interior point. For two-point grids (Adam betas, gradient clip), we follow standard LLM fine-tuning practice, as these ranges cover nearly all practically useful settings. All baselines share the same fixed training configuration (batch size, max sequence length, update steps). Replay baselines differ only by the replay data source.

Table 7: Hyperparameter sweep ranges and selected values for all baselines.

| Tunable hyperparameters (swept) | | | Fixed training settings | |
|---|---|---|---|---|
| **Hyperparameter** | **Sweep values** | **Selected** | **Setting** | **Value** |
| Learning rate | $\{3\times10^{-6}, 1\times10^{-6}, 5\times10^{-7}\}$ | $\mathbf{1\times10^{-6}}$ | Global batch size | **128** |
| Weight decay | $\{0.01, 0.001\}$ | **0.01** | Max sequence length | **8192** |
| Gradient clip | $\{0.5, 1.0\}$ | **1.0** | Update steps | **150** |
| Adam betas | $\{(0.9, 0.95), (0.9, 0.98)\}$ | $(\mathbf{0.9}, \mathbf{0.95})$ | Finetuning mode | full-parameter |

# F  ADDITIONAL CLARIFICATIONS

**CT and NL2SQL as evaluation workloads.** The CT and NL2SQL datasets used in our experiments are not intended as canonical OOD benchmarks. Instead, they represent the types of domain-shifted workloads that arise in practical post-training pipelines, where the target distribution differs substantially from the general-purpose pretraining corpus. Our goal is therefore to study continual-adaptation techniques under realistic conditions in which domain specialization can impact general capabilities. While extending to additional model families (e.g., Llama, Mistral) would further validate generality, we leave this for future work.

**Meaning of the removed epsilon.** Here, the removed $\epsilon$ refers to the standard AdamW denominator constant added inside the square root. MoClip still uses the usual learning rate schedule; only the $\epsilon$-based safety term is eliminated because the $\arctan 2$ formulation ensures bounded updates.

**Staleness of Stage 0 logits.** Although Stage 0 logits are collected before fine-tuning, they remain effective anchors in Stage 1. The goal of replay is not to approximate the current model but to preserve the predictive structure of the base model. Using static logits is analogous to fixed-teacher distillation and avoids the cost of repeatedly querying a frozen teacher. Empirically, we find that dynamic Top-$K$ replay maintains high retention even though the logits are collected only once.

