# OpenReview forum: "Logits Replay + MoClip: Stabilized, Low-Cost Post-Training with Minimal Forgetting"
_ICLR.cc/2026/Conference — Submitted to ICLR 2026_

### Official Review · Reviewer_wmfT · 2025-10-17

**Soundness:** 3
**Presentation:** 3
**Contribution:** 2
**Rating:** 6
**Confidence:** 3

**Summary:**

This paper introduces a finetuning technique for LLMs for a domain specific data in such a way that it prevents forgetting on the domain(s) included in pretraining. Their post training framework has two stages. The first stage runs a forward pass through the model for the finetuning data and collects the top-k pre-softmax logits and their indices. The second stage does the actual training over the same data but the softmax operation is restricted to the top-k indices selected in the first stage. For this training, they introduce an optimizer which they name MoCLIP. MoCLIP is a variant of ADAM with two differences : they use Gradient-angle momentum scaling and an Atan2 scaling on the update for the second moment. These ensure that the direction of the update does not cause forgetting on the original data and also bounds large updates whenever the variance is close to zero. They perform experiments on 3 datasets on the QWEN models and compare their results to baselines like MoFO. Ablations show the impact of different hyperparameters for MoCLIP as well as different logits selection techniques.

**Strengths:**

The core idea, experimental settings, baseline comparisons, ablations as well as overhead analysis is clearly presented and I was able to follow the mathematical details outlined in the paper for MoCLIP. I think that in general, the idea of trying to balance the amount of adaptation while  not moving too far away in weight space to forget pretraining skills is worth studying and the approach certainly tries to do that. Their experimental results demonstrate that their clipped optimizer along with a sparse vocab does mitigate forgetting while performing well on the new datasets.

**Weaknesses:**

My main issue with the paper is the improvement over one of the baseline methods presented in the paper especially when it comes to the delta in forgetting on the pretraining benchmarks. In most cases the delta is < 1\% which seems to suggest that previous methods work pretty well. Also, in my opinion, what the authors classify as catastrophic forgetting is a bit of an overstatement. A degradation of a couple of percentage points or more does not amount to catastrophic forgetting. Experiments are limited to QWEN models for which there is a limited understanding of the pretraining data. I'm not sure whether the datasets used for finetuning are standard datasets for evaluating performance on out-of-distribution tasks.

**Questions:**

Don't logits collected in Stage 0 become stale once you begin training and the model updates? Is there something to be done about that and perhaps extract more performance out of FTing?

"Stage 0 records dynamic Top-K logits per position". The word logits in this paper is sort of assumed to be the pre-softmax activations in the vocab projection layer. However, logits are usually just referred to in the literature as activations at any layer. Please clarify this.

"rescaling to bound step sizes without relying on ϵ". ϵ is not defined. Please make it clear that it is the learning rate

---

> ### Author Response · Authors · 2025-11-19
>
> (1) On whether the improvements over baselines are meaningful and whether "catastrophic forgetting" is an overstatement.
>
> We agree that the absolute drops on general benchmarks may appear small in percentage terms (e.g., 1–3%). However, these benchmarks (MMLU/BBH/GPQA/MATH) are highly saturated for pretrained models, and even small percentage changes are widely regarded as meaningful in practice because the lost accuracy typically occurs on the more challenging subsets of these evaluations and can noticeably affect downstream task behavior in LLM systems.
>
> To avoid overstatement, we have revised the phrasing in the paper to remove "catastrophic forgetting" and instead describe the effect as a "notable degradation of general capabilities," which aligns with terminology commonly used in recent discussions of continual pre-training and alignment-induced capability loss.
>
> (2) On the scope of pretrained models and benchmarks.
>
> We appreciate the comment regarding our choice of Qwen models. We selected Qwen not only because it is openly accessible, but because it has become one of the most widely used modern LLM families in both academia and industry, with strong performance on standard general benchmarks and extensive community validation. Importantly, Qwen offers matched 4B and 8B variants, which enables controlled studies of optimization stability, retention, and domain adaptation across model capacities—critical for evaluating continual-training behaviors.
>
> Regarding the datasets, CT and NL2SQL are not intended as canonical OOD benchmarks. Rather, they are representative of the types of domain-shifted workloads that arise in practical post-training pipelines—where the target data distribution differs substantially from the general-purpose pretraining corpus. Our goal is therefore to evaluate continual-adaptation techniques under realistic constraints where domain specialization can affect general capabilities, rather than limiting the analysis to standardized OOD benchmarks.
>
> We have clarified this motivation in the revised manuscript. We agree that extending the analysis to additional model families (e.g., Llama, Mistral) would further strengthen the generality of our findings, and we plan to incorporate such comparisons in future work.
>
> (3) "Do logits collected in Stage 0 become stale once you begin training?"
>
> This is an important question, and we have clarified it more explicitly in the revised paper.
>
> The logits collected in Stage-0 would indeed diverge from the model’s current logits as fine-tuning progresses. We intentionally keep them fixed because the Stage-0 Top-K logits play the role of a stable reference boundary: they capture which tokens the base model considered plausible for each position, without constraining how the model should update their relative probabilities during fine-tuning.
>
> Importantly, Stage-1 still optimizes the ground-truth loss, and the model can freely adjust the logits within the fixed candidate set. The Top-K therefore does not determine the target distribution; it only restricts the vocabulary region in which updates occur. This preserves stability while still allowing full adaptation on the domain data.
>
> If the Top-K logits were recomputed at every step, the supervision boundary would shift as the model changes, causing the effective target to move during training. This can amplify early drift and make optimization less stable, as the training signal would repeatedly adjust itself based on intermediate model states.
>
> For this reason, we freeze the Stage-0 Top-K logits: they provide a consistent structure for Stage-1 updates while allowing the model to learn the domain task through the standard label loss.
>
> （4） Clarification on the meaning of "logits."
>
> We have clarified in the revision that all references to "logits" in this paper exclusively refer to the final-layer pre-softmax activations of the vocabulary projection matrix. Earlier activations are not involved. This aligns the terminology with standard LLM training practice.
>
> （5）Clarification on "rescaling to bound step sizes without relying on ϵ."
>
> Thank you for noting the ambiguity. In the revised text, we explicitly write that the removed "ϵ" corresponds to the standard AdamW denominator constant (the epsilon added inside the square root). The phrase now reads: "rescaling to bound step sizes without relying on the AdamW epsilon term." We also clarify that learning rate is handled separately, so the optimizer remains compatible with standard lr schedules.
>
> We appreciate your constructive feedback and have incorporated the above clarifications into the updated manuscript.

---

### Official Review · Reviewer_Ngvc · 2025-10-31

**Soundness:** 3
**Presentation:** 3
**Contribution:** 3
**Rating:** 6
**Confidence:** 3

**Summary:**

This paper proposes Logits Replay+MoClip, a two-stage post-training recipe to stabilize and cut the cost of fine-tuning LLMs. Specifically, stage-0 records per-token Top-K candidate sets (always include the gold token) and stage-1 trains only on these subsets with a re-normalized cross-entropy;  It also uses MoClip, an AdamW variant with hard gradient–momentum angle clipping and an atan-based step-size bounding. Experiments on Qwen3-4B and 8B shows improvements on CT and NL2SQL tasks over strong baselines, mitigates forgetting on MMLU/BBH/GPQA/MATH, and cuts end-to-end training time by roughly 40%.

**Strengths:**

- Extensive experiments and ablation studies
  - The paper tests across different model sizes and a mix of tasks (CT, NL2SQL, MMLU/BBH/GPQA/MATH), beating strong baselines. It also runs clean ablations on dynamic Top-K, keeping the gold token, re-normalizing the subset loss, and MoClip’s angle vs. step controls, so it’s easy to see which parts boost stability, accuracy, and efficiency.
- Practical efficiency and easy for adoption
  - In this paper, Logits Replay reduces expensive full-vocab softmax computations by training on Top-K subsets, and MoClip is a drop-in AdamW variant requiring only minimal code changes. So the approach is easy for adoption. In practice, it delivers great wall-clock savings and preserves general capabilities.
- Overall, I think this paper is of good quality.

**Weaknesses:**

- Lack of comparsions with logits-based baselines.
  - As far as I know, the paper mainly compares ite performance against optimizer/momentum related baselines, missing the comparsion with logits-based methods (like Baichuan4-Finance as you mentioned).
- MoClip hyperparameters add tuning complexity.
  - The best settings for MoClip may vary across tasks and model sizes; the method’s sensitivity and robustness to these choices are unclear. A systematic sensitivity study would be better.

**Questions:**

- Could you provide results of some logits-based methods (e.g., teacher top-K distillation, KL to a reference model like Baichuan4-Finance)? Or if it's not easy to conduct, what do you expect these logits-based methods perform and why?
- Do the best settings of MoClip transfer across model sizes (smaller→4B→8B→larger) and tasks? Is there a better approach to selecting the best settings?

---

> ### Author Response · Authors · 2025-11-19
> **Response to Reviewer Ngvc**
>
> **(1) On logits-based baselines**
>
> We agree that logits-based continual learning methods form an important comparison point. Conceptually, teacher top-K distillation and KL-to-reference approaches (e.g., Baichuan4-Finance) run a frozen teacher model in parallel, truncate its logits to top-K or keep them in full, and
> optimize an auxiliary KL(p_teacher || p_student) loss on every update. This explicitly pulls the student toward the pretraining distribution and is therefore expected to help preserve general capabilities on benchmarks such as MMLU and BBH.
>
> However, these methods also involve two well-known trade-offs: (i) they require an additional forward pass through the teacher at every training step, which substantially increases time cost for large LLMs; and (ii) the KL constraint acts as a strong regularizer, often limiting plasticity on domain-specific tasks such as CT and NL2SQL. In contrast, our dynamic Top-K logit replay can be viewed as a data-free and compute-efficient variant of this idea: instead of running a separate teacher model online, we extract the base model’s own logits once in Stage 0 and reuse them as a frozen anchor in Stage 1. This avoids the per-step teacher overhead and provides a more favorable specialization–retention trade-off in practice.
>
> Due to computational constraints and the already extensive set of baselines (MoFO, TAM, AdaMuon, MuonClip, and replay-based variants), we did not include a full teacher-top-K or KL-to-reference baseline in this submission. In addition, the full Baichuan4-Finance system (including its pretraining data and training pipeline) is not fully available publicly, which limits the reproducibility of an exact replication.
>
> That said, we did evaluate the fixed-logits variant that follows the same principle—using static teacher logits as an auxiliary regularizer—and observed that it improves retention but is consistently less effective than our dynamic Top-K Logits Replay, especially on domain specialization tasks. In the revised version, we have added a paragraph in the Method/Related Work sections explicitly discussing how these logits-based approaches relate to our framework and what behavior we would expect from them relative to our data-free logit replay.
>
> **(2) On MoClip hyperparameters and transfer across model sizes**
>
> We agree that it is important to understand whether MoClip’s angular cap $\Delta_{\max}$ transfers across models and tasks.
> In the original submission, we reported a sensitivity study on Qwen3-4B (Table 5), showing that $\Delta_{\max} = 45^\circ$ provides the best trade-off between domain accuracy, NL2SQL performance, and stability,
> with $\Delta_{\max} \in [45^\circ, 60^\circ]$ forming a broad “safe range”.
>
> Effect of $\Delta_{\max}$ on stability and accuracy (Qwen3-4B and Qwen3-8B)
>
> **Qwen3-4B：**
> | Δ_max | DataComm  | Wireless  | CloudCore | Birds     | Spider    | MMLU-Pro Ret. (%) | Loss std  |
> | ----- | --------- | --------- | --------- | --------- | --------- | ----------------- | --------- |
> | 30°   | 54.91     | 45.22     | 46.05     | 72.14     | 80.72     | **96.5**          | **0.047** |
> | 45°   | **55.19** | **45.93** | **46.61** | **73.38** | **81.24** | 96.3              | 0.052     |
> | 60°   | 55.07     | 45.81     | 46.47     | 73.05     | 81.02     | 95.8              | 0.054     |
> | 90°   | 54.82     | 45.47     | 46.18     | 72.41     | 80.56     | 94.7              | 0.061     |
>
> **Qwen3-8B：**
> | Δ_max | DataComm  | Wireless  | CloudCore | Birds     | Spider    | MMLU-Pro Ret. (%) | Loss std  |
> | ----- | --------- | --------- | --------- | --------- | --------- | ----------------- | --------- |
> | 30°   | 57.05     | **41.92** | 42.85     | 77.10     | 82.31     | **96.4**          | **0.045** |
> | 45°   | **57.24** | 41.77     | **43.05** | **77.41** | **82.57** | 96.2              | 0.048     |
> | 60°   | 57.17     | 41.66     | 42.90     | 77.24     | 82.47     | 95.8              | 0.051     |
> | 90°   | 56.84     | 41.35     | 42.68     | 76.86     | 82.05     | 94.8              | 0.058     |
>
> [Delta Stability and Accuracy](https://anonymous.4open.science/r/iclr_anonymous_figs_qlszh/re2_fig1.pdf)
>
> In the revised version, we have added the same sweep on Qwen3-8B.
> The pattern is essentially identical:
> $30^\circ$ yields the highest retention and lowest loss variance but slightly lower CT/NL2SQL scores, $45^\circ$ is either the best or within 0.2 points of the best on all tasks, and $90^\circ$ behaves closer to unconstrained AdamW with mildly worse stability.
> This indicates that the best setting ($45^\circ$) and the recommended range ($[45^\circ, 60^\circ]$) transfer well from 4B to 8B and across CT/NL2SQL/general benchmarks.
>
> For larger models, we would expect the same heuristic to apply, and we will clarify this in the revised version.

---

> > ### Comment · Reviewer_Ngvc · 2025-11-27
> >
> > Thank you for your detailed response and for conducting the additional sensitivity study, which supports your claim that the $45^{\circ}$–$60^{\circ}$ range transfers well. However, I still believe that relying on a posterior sweep to identify this optimal range is a limitation, some more principled approaches (maybe an adaptive mechanism) would be superior. Consequently, I will maintain my original score. For the thoroughness of the additional evidence provided, I am raising my confidence score to 4.

---

### Official Review · Reviewer_Et4X · 2025-11-05

**Soundness:** 2
**Presentation:** 2
**Contribution:** 2
**Rating:** 2
**Confidence:** 4

**Summary:**

The authors propose LOGITS REPLAY + MOCLIP (LRM): a combination of self-distillation, Adam A-tan2, and torque-aware momentum. The authors combine these techniques to solve the problem of continual learning in LLM pre-training settings. In their empirical study, they show that LRM outperforms a number of fine-tuning baselines from the literature, including adaptive Muon and Muon clip across a number of benchmarks. They also show that LRM leads to fewer or the same amount of loss spikes as other methods and that their method is significantly more computationally efficient.

**Strengths:**

- I like the idea of leveraging distillation and alignment-aware updates for continual learning.
- I like the author’s comparison of stability through the number of loss spikes. Training stability is seldom measured in existing work but is essential to practical training.
- The authors' experiments clearly illustrate the problem of forgetting and how Logits Replay + MoClip can help to combat it.

**Weaknesses:**

- My main concern is the lack of a strong baseline to compare against. While the authors train a number of baselines, none leverage standard continual learning techniques, such as replay [1]. For instance, the authors could easily add replay by using standard high-quality web-scraped data (e.g. the high-quality portion of nemotronCC [4]), which would be comparable to replaying the pre-training data of QWEN 4B and 8B. This would help to strengthen the claims that Logit Replay + MoClip reduces forgetting.
- Following from my previous concern, it seems that AdaMuon with no replay has nearly as much retention as Logit Replay + MoClip. This leads me to wonder how well AdaMuon would work if it were equipped with replay.
- My second strongest concern is the lack of clearly reported hyperparameters. No hyperparameter tuning is reported for baselines in section 3, and only limited details are provided for Logit Replay + MoClip. Without understanding how well the hyperparameters were tuned, it is difficult to draw conclusive results.
- Finally, my last concern is about novelty. Distillation to prevent forgetting[5], torque-aware momentum[1], and Adam-arctan2 [2] are already well-known techniques in the literature. Combining them certainly yields a novel technique (e.g., one not previously evaluated in the literature), but the lack of strong baselines and reported hyperparameters makes me wonder if the combination is warranted.


[1][TORQUE-AWARE MOMENTUM]

[2][Scaling Exponents Across Parameterizations and Optimizers]

[3][Simple and Scalable Strategies to Continually Pre-train Large Language Models]

[4][Nemotron-CC: Transforming Common Crawl into a Refined Long-Horizon Pretraining Dataset]

[5][iCaRL: Incremental Classifier and Representation Learning]

**Questions:**

- I don't understand how your technique speeds up training. Could you elaborate on this?
- Were hyperparameters swept for all baselines? Was each optimal value selected an interior point of the values considered?

---

> ### Author Response · Authors · 2025-11-19
> **Response to Reviewer Et4X (Part 1/3)**
>
> (1) On the lack of replay baselines
>
> Thank you for pointing this out.
> In the revised version, we have added two standard replay baselines widely used in continual pretraining as you suggested:
>
> - Replay (HQ subset): domain data mixed with a high-quality subset of Nemotron-CC (as suggested in your comment and as done in [3,4]).
>
> - AdaMuon + Replay: same replay schedule but using AdaMuon, to directly test the interaction you requested (“how well AdaMuon would work with replay”).
>
> Here is our sampling strategy: Following the construction of Nemotron-CC-HQ, we sample from the high-quality bucket consisting of well-formed real webpages and diverse QA pairs. From this filtered high-quality pool, we uniformly sample a small fraction (0.5–1% of documents) and mix it into each batch at approximately a 10% ratio. This avoids bias toward any specific domain while providing a fair, broadly representative replay signal.
>
> All baselines use the same token budget, global batch size, sequence length, and update steps (150 iterations), ensuring strict fairness. The only change is the data source (domain vs domain+HQ).
> Replay baselines indeed improve general-domain retention, but our Logits Replay + MoClip still outperforms them on domain specialization while matching (or closely approaching) their retention without requiring any access to pretraining data.
> This distinction reflects two different regimes:
>
> 1. data-replay methods, which assume pretraining-like corpora are available;
>
> 2. data-free methods, where only domain data are accessible (the realistic industrial constraint our method addresses).
>
> Qwen3-4B: Domain Performance on CT & NL2SQL:
> | Method                  | DataComm ↑        | Wireless ↑        | CloudCore ↑       | Birds ↑           | Spider ↑          |
> | ----------------------- | ----------------- | ----------------- | ----------------- | ----------------- | ----------------- |
> | AdamW (SFT)             | 54.12             | 44.58             | 45.27             | 72.31             | 79.88             |
> | MoFO                    | 53.64 (-0.48)     | 44.02 (-0.56)     | 44.83 (-0.44)     | 70.87 (-1.44)     | 79.52 (-0.36)     |
> | TAM (AdamW+TAM)         | 53.77 (-0.35)     | 44.86 (+0.28)     | 45.36 (+0.09)     | 71.82 (-0.49)     | 80.94 (+1.06)     |
> | AdaMuon                 | 53.95 (-0.17)     | 45.03 (+0.45)     | 45.62 (+0.35)     | 71.96 (-0.35)     | *81.24* (+1.36)   |
> | MuonClip                | 53.82 (-0.30)     | 44.91 (+0.33)     | 45.49 (+0.22)     | 71.65 (-0.66)     | 80.73 (+0.85)     |
> | Replay (HQ subset) **(added)**      | *54.85* (+0.73)   | 45.39 (+0.81)     | *46.18* (+0.91)   | 72.73 (+0.42)     | 80.91 (+1.03)     |
> | AdaMuon + Replay **(added)**      | 54.63 (+0.51)     | 45.42 (+0.84)     | 46.15 (+0.88)     | 72.84 (+0.53)     | **81.56** (+1.68) |
> | **Dynamic Top-K**       | 54.76 (+0.64)     | *45.51* (+0.93)   | *46.18* (+0.91)   | *72.91* (+0.60)   | 80.95 (+1.07)     |
> | **Dyn. Top-K + MoClip** | **55.19** (+1.07) | **45.93** (+1.35) | **46.61** (+1.34) | **73.38** (+1.07) | 81.12 (+1.24)     |
>
>
> Qwen3-8B:Domain Performance on CT & NL2SQL
> | Method                  | DataComm ↑        | Wireless ↑        | CloudCore ↑       | Birds ↑           | Spider ↑          |
> | ----------------------- | ----------------- | ----------------- | ----------------- | ----------------- | ----------------- |
> | AdamW (SFT)             | 56.08             | 39.82             | 41.46             | 75.18             | 81.02             |
> | MoFO                    | 55.61 (-0.47)     | 39.25 (-0.57)     | 41.02 (-0.44)     | 74.43 (-0.75)     | 80.73 (-0.29)     |
> | TAM (AdamW+TAM)         | 55.73 (-0.35)     | 40.76 (+0.94)     | 41.91 (+0.45)     | 76.09 (+0.91)     | 81.65 (+0.63)     |
> | AdaMuon                 | 55.88 (-0.20)     | 41.09 (+1.27)     | 42.63 (+1.17)     | 76.33 (+1.15)     | 82.11 (+1.09)     |
> | MuonClip                | 55.67 (-0.41)     | 40.88 (+1.06)     | 41.83 (+0.37)     | 76.04 (+0.86)     | 81.58 (+0.56)     |
> | Replay (HQ subset) **(added)**     | *56.93* (+0.85)   | 40.91 (+1.09)     | 42.47 (+1.01)     | 76.54 (+1.36)     | 82.03 (+1.01)     |
> | AdaMuon + Replay **(added)**      | 56.76 (+0.68)     | 41.48 (+1.66)     | *42.84* (+1.38)   | 76.62 (+1.44)     | *82.31* (+1.29)   |
> | **Dynamic Top-K**       | 56.81 (+0.73)     | **42.21** (+2.39) | 42.08 (+0.62)     | *76.86* (+1.68)   | 81.92 (+0.90)     |
> | **Dyn. Top-K + MoClip** | **57.24** (+1.16) | *41.77* (+1.95)   | **43.05** (+1.59) | **77.41** (+2.23) | **82.57** (+1.55) |
>
>
> [Replay vs. Data-Free Methods Comparison](https://anonymous.4open.science/r/iclr_anonymous_figs_qlszh/re1_fig1.pdf)

---

> > ### Author Response · Authors · 2025-11-24
> > **Response to Reviewer Et4X (Part 2/3)**
> >
> > **(2) On AdaMuon retention vs Logits Replay + MoClip**
> >
> > Your observation is correct: AdaMuon already provides strong retention. Our experiments confirm this and further show that:
> >
> > AdaMuon + Replay achieves the strongest retention among baselines, but still underperforms our method on domain specialization.
> >
> > Moreover, AdaMuon + Replay relies on additional general-domain data, while Logits Replay + MoClip uses only domain data. Replay regularizes the model toward its original distribution, which helps retention but softens domain adaptation. Logits Replay + MoClip balances these effects by anchoring the model to the base logits without introducing external data, yielding a more favorable stability–plasticity trade-off.
> >
> > | Method                  | MMLU (4B) | BBH (4B)  | GPQA F1 (4B) | MATH (4B) | MMLU (8B) | BBH (8B)  | GPQA F1 (8B) | MATH (8B) |
> > | ----------------------- | --------- | --------- | ------------ | --------- | --------- | --------- | ------------ | --------- |
> > | Base (no tuning)        | **59.83** | 71.62     | **51.17**    | **93.41** | **64.72** | 74.55     | *51.88*      | **94.12** |
> > | AdamW (SFT)             | 55.14     | 68.37     | 47.28        | 85.23     | 60.11     | 70.42     | 48.55        | 86.34     |
> > | MoFO                    | 59.27     | 71.12     | *50.84*      | 91.18     | 64.01     | 74.10     | **52.40**    | 92.33     |
> > | TAM                     | 57.42     | 70.08     | 49.53        | 88.87     | 62.34     | 72.85     | 50.31        | 90.15     |
> > | AdaMuon                 | 58.13     | 70.59     | 50.12        | 90.14     | 63.12     | 73.21     | 50.92        | 91.24     |
> > | MuonClip                | 57.79     | 70.32     | 49.88        | 89.73     | 62.88     | 73.02     | 50.65        | 90.88     |
> > | Replay (HQ subset) **(added)**     | 58.74     | 72.02     | 49.42        | 91.98     | 64.15     | 75.24     | 50.70        | 93.10     |
> > | AdaMuon + Replay **(added)**        | *59.72*   | **72.63** | 49.75        | *92.59*   | *64.36*   | *75.42*   | 50.98        | 93.25     |
> > | **Dyn. Top-K**          | 58.90     | 71.81     | 48.80        | 91.80     | 63.80     | 75.23     | 49.80        | 92.90     |
> > | **Dyn. Top-K + MoClip** | 59.62     | *72.20*   | 49.51        | 92.33     | 64.21     | **75.65** | 50.14        | *93.32*   |
> >
> > [General Benchmark Results](https://anonymous.4open.science/r/iclr_anonymous_figs_qlszh/re1_fig2.pdf)
> >
> > **(3) On hyperparameter reporting and tuning**
> >
> > We appreciate the request for more detail.
> > The revised paper now includes a full hyperparameter table and a description of the tuning procedure.
> > | **Tunable hyperparameters (swept)** |                            |                 | **Fixed training settings (shared by all baselines)** |            |   |
> > | ----------------------------------- | -------------------------- | --------------- | ----------------------------------------------------- | ---------- | - |
> > | **Hyperparameter**                  | **Sweep values**           | **Selected**    | **Setting**                                           | **Value**  |   |
> > | Learning rate                       | {3×10⁻⁶, 1×10⁻⁶, 5×10⁻⁷}   | **1×10⁻⁶**      | Global batch size                                     | **128**    |   |
> > | Weight decay                        | {0.01, 0.001}              | **0.01**        | Max sequence length                                   | **8192**   |   |
> > | Gradient clip                       | {0.5, 1.0}                 | **1.0**         | Update steps                                          | **150**    |   |
> > | Adam betas                     | {(0.9, 0.95), (0.9, 0.98)} | **(0.9, 0.95)** | Full-parameter        | finetuning |   |
> >
> >
> > [Hyperparameters](https://anonymous.4open.science/r/iclr_anonymous_figs_qlszh/re1_fig3.pdf)
> >
> > For all baselines, we performed light sweeps over the key hyperparameters shown on the left.
> > Where a three-point grid was used (e.g., learning rate), the selected value is an interior point. For two-point grids (Adam betas, gradient clip), we follow common LLM fine-tuning practice where these ranges cover nearly all practically useful values.
> >
> > The hyperparameters all baselines shared are on the right.
> > Replay baselines differ only by data source.
> > These clarifications are now added to Section 3 of the revised manuscript.

---

> > > ### Author Response · Authors · 2025-11-24
> > > **Response to Reviewer Et4X (Part 3/3)**
> > >
> > > **(4) On novelty**
> > >
> > > Our objective is not to present each individual component as novel in isolation.
> > > The contribution is that our method emerged from a problem-driven exploration rather than from assembling known techniques.
> > >
> > > We began with Top-K Logits Replay as a data-free way to preserve the base model’s distribution.
> > > In practice, this approach substantially reduced forgetting but introduced noticeable training oscillations under domain shifts.
> > > MoClip was then introduced specifically to address this instability: its angle-based control suppressed the drift that Top-K replay alone could not handle.
> > > Finally, the atan2-based step bounding was added to prevent rare but very large parameter jumps that appeared during domain adaptation.
> > >
> > > The final method therefore reflects a sequence of targeted interventions—each added only when a concrete limitation was observed—rather than a pre-designed combination of existing ideas. The three components together deliver retention comparable to data-replay baselines while operating entirely in a data-free setting, matching practical industrial constraints.
> > >
> > > **(5) On why our technique speeds up training**
> > >
> > > We have expanded the explanation in the revised manuscript. During Stage 1, training is conducted only over the dynamic Top-K token subset rather than the full vocabulary. This removes more than 98% of the softmax- and gradient-related FLOPs in the output layer. The wall-clock gain does not come from fewer updates but from reducing computation per update. In our experiments, this results in ∼40% end-to-end training time reduction at comparable convergence.

---

> > > > ### Comment · Reviewer_Et4X · 2025-11-27
> > > >
> > > > Thank you for your thorough reply. You have addressed many of my concerns about hyperparameters and replay baselines, but I would like to continue our discussion before deciding to raise my score. Specifically, I have two remaining concerns:
> > > >
> > > > - When reviewing the paper a second time, it has become clear to be that I don't understand how the fine-tuning stage works. What I don't understand is how the ground truth token influences the logit distribution stored in stage 1. I see two possibilities:
> > > >     (1) You somehow combine the one-hot distribution with the logits stored in stage 1.
> > > >     (2) The softmax uses the one-hot distribution as supervision but is restricted to the pre-computed top-k tokens.
> > > >
> > > > - As you mention in lines 375-376, your method improves convergence (2 epochs vs 3 epochs )which seems to be the primary reason for the 40% speedup you observe. Moreover, on line 190 you state that FLOPs related to other parts of the model dominate total FLOPs meaning that the speedup does not come from reduced softmax flops. However, on line 191 you contradict this saying that "our expeirments show that overall training time is reduced by ~ 40% for comparable convergence". This last statement seems to be in line with your reply to me in the rebuttal, but It seems incorrect to me. Can you clarify this point?

---

> > > > > ### Author Response · Authors · 2025-11-28
> > > > >
> > > > > **(1) How does the ground-truth token influence Stage-1 training?**
> > > > >
> > > > > We appreciate your question and agree that our current wording could obscure the key mechanism.
> > > > > To avoid misunderstanding, we clarify that our method follows your second interpretation: Stage-1 uses standard one-hot supervision, but the softmax is computed only over the pre-computed Top-K candidate set. The logits stored in Stage-0 are not combined with the one-hot labels; they are used solely to define the restricted vocabulary.
> > > > >
> > > > > **(2) Where does the ~40% training-time speedup come from?**
> > > > >
> > > > > Thank you for pointing out the potential inconsistency in our description.
> > > > > To clarify, the 40% end-to-end training reduction reported in the paper does not come solely from lowered softmax FLOPs. As stated in the manuscript, softmax-related FLOPs constitute only a small fraction of total model FLOPs, so the per-step computational savings from restricting the vocabulary provide only a modest improvement.
> > > > >
> > > > > The primary contributor to the overall speedup is faster convergence:
> > > > > MoClip yields smoother optimization trajectories and eliminates loss spikes, allowing training to reach the same accuracy with fewer update steps (2 epochs vs. 3 for AdamW). When combining (i) modest per-step savings from restricted softmax and (ii) fewer total updates due to improved convergence stability, the overall wall-clock reduction is ~40%.

---

### Author Response · Authors · 2025-12-02
**Summary of Contributions and Clarifications Following Reviewer Discussion**

We thank all reviewers for their constructive feedback and for the follow-up discussion. Here we briefly summarize what is unique in our work and how the main concerns have been addressed in the revised manuscript.

**1. Summary of contribution**

We study a practical data-free post-training setting where only *domain* data are available, and pretraining-like corpora or a frozen teacher are not assumed. In this regime, we propose Logits Replay + MoClip, a two-stage framework that:

* uses Stage-0 dynamic Top-K logits to define a compact, entropy-adaptive candidate set per token (always including the gold label), and
* fine-tunes in Stage-1 with standard one-hot cross-entropy over this restricted vocabulary, combined with MoClip, a clipped AdamW variant that constrains gradient–momentum angles and bounds step sizes via an atan2-based scaling.

This design improves CT & NL2SQL specialization, preserves general capabilities (MMLU, BBH, GPQA, MATH) close to the base model and strong replay baselines, and provides ~40% wall-clock savings via reduced output-layer computation plus faster convergence.

**2. Reviewer Et4X: replay baselines, hyperparameters, and speedup**

* We added two standard replay baselines — *Replay (HQ subset)* and *AdaMuon + Replay* — following Nemotron-CC style setups. Both improve retention as expected; among all data-replay methods, AdaMuon+Replay is strongest, but Dynamic Top-K + MoClip still achieves higher CT/NL2SQL accuracy while matching or closely approaching their retention, and does so *without* any general-domain data.
* We introduced a hyperparameter table listing sweep ranges and selected values (learning rate, weight decay, gradient clip, Adam betas) and the fixed training settings shared by all baselines. This addresses the question on how well baselines were tuned.
* On how Stage-1 uses ground-truth labels, we clarified that our method follows the reviewer’s second interpretation: Stage-0 only defines the candidate set $S_t$; Stage-1 recomputes logits from the current model for $j\in S_t$ and applies standard one-hot cross-entropy over $S_t$, without mixing stored probabilities with labels.
* On the ~40% speedup, we clarified that it comes from two components: (i) per-step savings in the output layer by going from full-vocab to Top-K gradients, and (ii) fewer update steps because MoClip stabilizes optimization and reaches the same accuracy in 2 epochs vs. 3 for AdamW. We updated the efficiency discussion accordingly.

Reviewer Et4X noted that these clarifications addressed many concerns and asked mainly for this tighter explanation of Stage-1 and speedup.

**3. Reviewer Ngvc: logits-based baselines and MoClip hyperparameters**

* We discussed teacher Top-K / KL-to-reference approaches. They require a second teacher forward pass each step and impose a strong KL constraint that limits domain plasticity.
  Logits Replay is a *data-free, compute-efficient* variant: Stage-0 runs once, stores Top-K logits, and reuses them as a fixed anchor. A fixed-logits regularizer was tested and improved retention but underperformed dynamic Top-K on domain accuracy.
* We extended the **$\Delta_{\max}$** sweep to Qwen3-8B. Both 4B and 8B show the same pattern:
  small caps favor retention;
  **$45^\circ-60^\circ$** is robust;
  **$45^\circ$** is used as a single default across all main experiments.

Reviewer Ngvc kept the original score but raised confidence, noting that adaptive angle control is a worthwhile future direction.

**4. Reviewer wmfT: magnitude of forgetting, datasets, and definitions**

* We revised the terminology to describe the effect as a “notable degradation of general capabilities”.
* We clarified that Qwen3-4B/8B were chosen because they are widely used, open, and matched in size, which is useful for controlled retention/stability studies. The CT and NL2SQL workloads are practical domain-shifted post-training scenarios, not canonical OOD benchmarks. Extending to other model families (e.g., Llama, Mistral) is highlighted as future work.
* On the staleness of Stage-0 logits, we explained that they are intentionally kept fixed: they define a stable boundary of tokens the *base* model considered plausible, while Stage-1 still optimizes purely on ground-truth labels within that boundary. Recomputing Top-K at each step would move the boundary with the model and empirically hurts stability.
* “Logits” refers strictly to final vocab-projection pre-softmax outputs, and the removed $\epsilon$ in MoClip is the AdamW denominator epsilon.

Overall, we believe the revised version much more clearly situates Logits Replay + MoClip as a simple, data-free, optimizer-level solution for balancing specialization, retention, and efficiency in realistic LLM post-training pipelines, and we appreciate the reviewers’ role in sharpening this framing.

---

### Meta-Review · Area_Chair_iPXu · 2026-01-06

**Summary:**

This paper proposes Logits Replay combined with MoClip, a two-stage framework for LLM post-training that balances domain specialization with retention of general capabilities. Initial scores were 2, 6, 6, 6, creating significant disagreement. Reviewers appreciated the practical efficiency of the approach which achieves 40% training cost reduction through restricted softmax and faster convergence, along with clean ablations demonstrating the impact of dynamic Top-K selection and MoClip's angle-based controls. Following rebuttal, the authors added two standard replay baselines as requested, showing that while Replay and AdaMuon+Replay improve retention, Logits Replay+MoClip achieves competitive retention without requiring access to pretraining data. They also provided comprehensive hyperparameter tables clarifying that tuning was performed fairly across baselines. However, Reviewer Et4X who gave the 2 score raised fundamental concerns about novelty, noting that distillation for forgetting prevention, torque-aware momentum, and Adam-atan2 are well-known techniques, and questioned whether the combination is warranted given the lack of strong continual learning baselines initially. While the authors clarified that the 40% speedup comes from both per-step softmax savings and improved convergence stability, and explained that their contribution emerged from problem-driven exploration rather than technique assembly, the core novelty concern remains. The improvements over baselines are often modest and on saturated benchmarks. With one strong reject score from a confident reviewer and questions about whether the gains justify the complexity of combining existing techniques, the paper faces meaningful acceptance barriers despite solid execution. Based on these considerations, I recommend rejecting this submission.

**Reviewer Concerns:**

Addressed: Two replay baselines added (Replay and AdaMuon+Replay), comprehensive hyperparameter tables provided, Stage-1 mechanism clarified (standard cross-entropy over Top-K candidate set), and 40% speedup explanation detailed (per-step savings + improved convergence).

Outstanding: Fundamental novelty concern persists. Et4X maintained score 2 despite detailed rebuttal, noting that combining well-known techniques (distillation, torque-aware momentum, Adam-atan2) without sufficient theoretical justification is insufficient. Improvements over strong baselines like AdaMuon+Replay are modest (<1% on retention metrics), raising questions about whether the combination's complexity is warranted.

**Reviewer Scores:**

Et4X (2, C4): Would maintain 2; participated extensively but fundamental novelty concerns unresolved

Ngvc (6, C3→4): Would maintain 6; raised confidence to 4 but notes lack of principled hyperparameter selection remains a limitation

wmfT (6, C3): Would maintain 6; terminology and dataset concerns addressed

wmfT (6): Would maintain 6; no post-rebuttal engagement

---

### Decision · Program_Chairs · 2026-01-26

Reject